# Speeding up NAS with Adaptive Subset Selection

**Vishak Prasad C.**[1] **Colin White**[2] **Sibasis Nayak**[1] **Paarth Jain**[1] **Aziz Shameem**[1]
**Prateek Garg**[1] **Ganesh Ramakrishnan**[1]

[1]Indian Institute Of Technology, Bombay
[2]Abacus.AI

**Abstract** A majority of recent developments in neural architecture search (NAS) have been aimed at decreasing the computational cost of various techniques without affecting their final performance. Towards this goal, several low-fidelity and performance prediction methods have been considered, including those that train only on subsets of the training data. In this work, we present an adaptive subset selection approach to NAS and present it as complementary to state-of-the-art NAS approaches. We uncover a natural connection between one-shot NAS algorithms and adaptive subset selection and devise an algorithm that makes use of state-of-the-art techniques from both areas. We use these techniques to substantially reduce the runtime of DARTS-PT (a leading one-shot NAS algorithm) with minimal sacrifice to accuracy and to give reasonable runtime reductions for BOHB and DEHB (leading multi-fidelity optimization algorithms) as well. In experiments, we find architectures on CIFAR-10 that give 5% increase in performance over DARTS-PT while reducing the time required by more than a factor of 8. Our results are consistent across multiple datasets, and towards full reproducibility, we release all our code at `https://github.com/pcvishak/SubsetSelection_NAS`.

## 1 Introduction

Neural architecture search (NAS), the process of automating the design of high-performing neural architectures, has been used to discover architectures that outpace the best human-designed neural networks (Dai et al., 2020; Real et al., 2019; Tan and Le, 2019). Early NAS algorithms used black-box optimization methods such as reinforcement learning (Pham et al., 2018; Zoph and Le, 2017) and Bayesian optimization (Kandasamy et al., 2018). A majority of recent developments have focused on decreasing the cost of NAS without sacrificing performance.

Toward this direction, 'one-shot' methods improve the search efficiency by training just a single over-parameterized neural network (supernetwork) (Bender et al., 2018; Liu et al., 2019). For example, the popular DARTS (Liu et al., 2019) algorithm applies a continuous relaxation to the architecture parameters, allowing the architecture parameters and the weights to be simultaneously optimized via gradient descent. While many follow-up works have improved the performance of DARTS (Laube and Zell, 2019; Wang et al., 2021; Zela et al., 2020), the algorithms are still slow and require computational resources that are expensive in terms of budget and CO2 emissions (Tornede et al., 2021).

On the other hand, the field of subset selection for efficient machine learning-based model training has seen a flurry of activity. In this area of study, facility location (Mirzasoleiman et al., 2020), clustering (Clark et al., 2020), and other subset selection algorithms are used to select a representative subset of the training data, substantially reducing the runtime of model training. Recently, adaptive subset selection algorithms have been used to speed up model training even further (Killamsetty et al., 2021, 2020). Adaptive subset selection is a powerful technique which regularly updates the current subset of the data as the search progresses, to ensure that the performance of the model is maintained.

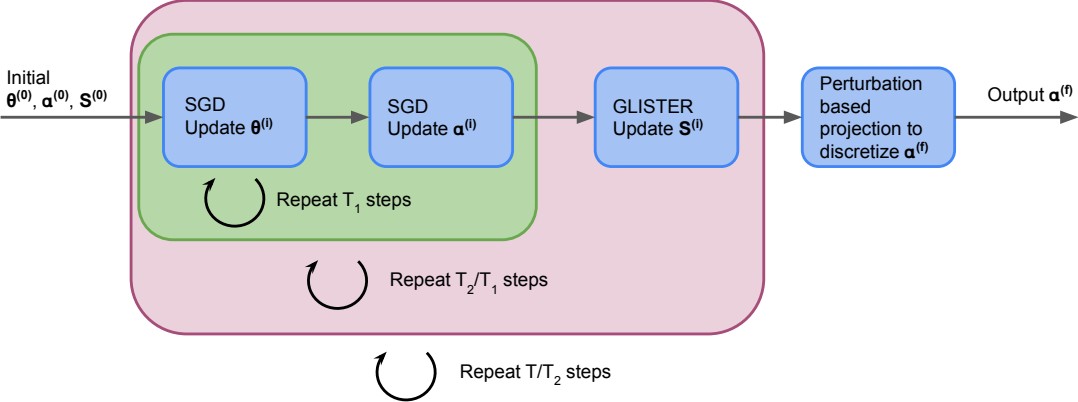

Figure 1: Overview of ADAPTIVE-DPT. The algorithm starts with the initial set of weights $\theta^{(0)}$, architecture-parameters $\alpha^{(0)}$, and subset of the training data $S^{(0)}$. Throughout the search, the weights $\theta^{(i)}$ and architecture-parameters $\alpha^{(i)}$ are updated with SGD, and the subset $S^{(i)}$ is updated with GLISTER, according to different time schedules. Then the final architecture $\alpha^{(f)}$ is discretized and returned.

In this work, we combine state-of-the-art techniques from both adaptive subset selection and NAS to devise new algorithms. First, we uncover a natural connection between one-shot NAS algorithms and adaptive subset selection: DARTS-PT (Wang et al., 2021) (a leading one-shot algorithm) and GLISTER, (Killamsetty et al., 2020) (a leading adaptive subset selection algorithm) are both cast as bi-level optimization problems on the training and validation sets, allowing us to formulate a combined approach, ADAPTIVE-DPT, as a mixed discrete and continuous bi-level optimization problem (see Figure 1 for an overview). Next, we also combine adaptive subset selection with BOHB (Falkner et al., 2018a) and DEHB (Awad et al., 2021), two leading multi-fidelity optimization approaches, to devise ADAPTIVE-BOHB and ADAPTIVE-DEHB, respectively.

Across several search spaces, we show that the resulting algorithms achieve a moderate to significant improvement in runtime, without sacrificing performance. Specifically, due to the use of adaptive subset selection, the training data can often be reduced to 10% of the full training set size, resulting in up to an order of magnitude decrease in runtime, with minimal reduction to accuracy.

To validate these approaches, we compare against baselines such as facility location, entropy-based subset selection (Na et al., 2021), and random subset selection. Facility location itself is a novel baseline for NAS applications; the codebase we release, which includes four different subset selection algorithms (*c.f.* Sec. 4) integrated into one-shot NAS, may be of independent interest.

**Our contributions**. We summarize our main contributions.

- We introduce ADAPTIVE-DPT, the first NAS algorithm to make use of adaptive subset selection. The training time needed to find high-performing architectures is substantially reduced. We also add facility location as a novel baseline for subset selection applied to NAS. (*c.f.* Sec. 3). We extend our idea to show that adaptive subset selection complements hyperparameter optimization algorithms in the form of ADAPTIVE-BOHB and ADAPTIVE-DEHB.

- Through extensive experiments, we show that ADAPTIVE-DPT, ADAPTIVE-BOHB and ADAPTIVE-DEHB often substantially reduce the runtime needed for running DARTS-PT, BOHB, and DEHB, respectively, with little to no decrease in the final (test) accuracy of the returned architecture (*c.f.* Sec. 4).

For reproducibility, we release all of our code.

## 2 Related Work

**Neural architecture search**. NAS has been studied since the 1980s (Angeline et al., 1994; Dress, 1987; Kitano, 1990; Tenorio and Lee, 1988) and has been revitalized in recent years (Liu et al., 2019; Zoph and Le, 2017). The initial set of approaches focused on evolutionary search (Real et al., 2019; Stanley and Miikkulainen, 2002), reinforcement learning (Pham et al., 2018; Zoph and Le, 2017), and Bayesian optimization (Kandasamy et al., 2018; White et al., 2021). More recent trends have focused on reducing the computational complexity of NAS using various approaches. One line of work aims to predict the performance of neural architectures before they are fully trained, through low-fidelity estimates such as training for fewer epochs (Ru et al., 2020; Zhou et al., 2020), learning curve extrapolation (Domhan et al., 2015; Yan et al., 2021), or 'zero-cost proxies' (Abdelfattah et al., 2021; Mellor et al., 2020).

Another line of work takes a *one-shot* approach by representing the entire space of neural architectures by a single 'supernetwork', and then performing gradient descent to efficiently converge to a high-performing architecture (Liu et al., 2019). Since the release of the original differentiable architecture search method (Liu et al., 2019), several follow-up works have attempted to improve its performance (Laube and Zell, 2019; Li et al., 2021; Liang et al., 2019; Xu et al., 2019; Zela et al., 2020). Recently, Wang et al. (2021) introduced a more reliable perturbation-based operation scoring technique while computationally returning the final architecture, yielding more accurate models compared to DARTS.

**Subset selection**. Several approaches have been developed in the field of subset selection for efficient model training. Popular fixed subset selection methods include coreset algorithms (Har-Peled and Mazumdar, 2004; Mirzasoleiman et al., 2020), facility location (Mirzasoleiman et al., 2020), and entropy-based methods (Na et al., 2021). Recently, Killamsetty et al. (2020) proposed GLISTER as an adaptive subset selection method based on a greedy search; an adaptive gradient-matching algorithm for subset selection was also subsequently proposed (Killamsetty et al., 2021).

**Subset selection in NAS**. A few existing works have applied fixed (non-adaptive) subset selection to the field of NAS. Na et al. (2021) consider three subset selection approaches: forgetting events, $k$-center, and entropy-based techniques, showing that the entropy-based approaches result in the best speedup in the case of DARTS. Shim et al. (2021) consider core-set sampling to speed up PC-DARTS by a factor of 8. Some more recent work (Killamsetty et al., 2022) employs subset selection algorithms to obtain greater speed-ups over multi-fidelity methods such as Hyperband (Li et al., 2018) and ASHA (Li et al., 2020). Another league of recent work uses a generative model to create a small set of *synthetic* training data, which in turn is used to efficiently train architectures during NAS (Rawal et al., 2020; Such et al., 2019).

Very recently, there has been more work on subset selection for NAS (Lu et al., 2024; Tu et al., 2023; Xie et al., 2024). We refer to Bartoldson et al. (2023); Lei and Tao (2023) for comprehensive surveys, and we discuss the most relevant works below. The most similar technique to our approach is perhaps Yao et al. (2023), recent subsequent work, which proposes a framework for bi-level adaptive subset selection for one-shot NAS. However, they use an mixture of objectives for subset selection rather than directly integrating the subset selection optimization into the existing architecture parameter optimization. Furthermore, they do not test their method with DARTS-PT (Wang et al., 2021). Finally, Jain et al. (2023) devise `SubSelNet`, a subset selection framework that uses a fast neural surrogate alongside an optimization routine in order to efficiently find the best subset of training data for new architectures. However, their method requires an initial surrogate model which uses additional computational resources.

## 3 Methodology

In this section, we first present the requisite background on supernet-based NAS algorithms and adaptive subset selection algorithms. Then, we introduce our approach, which combines these two techniques.

### 3.1 One-shot NAS Preliminaries

We begin by reviewing the ideas behind DARTS and DARTS-PT. The DARTS search space consists of cells, with each cell expressed as a directed acyclic graph, where each edge $(i, j)$ can take on choices of operations $o^{(i,j)}$ such as max_pool_3x3 or sep_conv_5x5. Let us denote the entire set of possible operations by $\mathcal{O}$. Each choice of operation for a given edge $(i, j)$, has a corresponding continuous valued variable $\alpha^{(i,j)}$. Let $\mathcal{U}$ and $\mathcal{V}$ denote the training and validation sets respectively. Further, let us denote the training and validation losses by $\mathcal{L}_{\text{train}}$ and $\mathcal{L}_{\text{val}}$, respectively. For any given dataset, these losses are a function of the architecture parameters and the architecture itself.

DARTS and DARTS-PT are both gradient-based optimization methods that train a supernetwork consisting of weights $\theta$ and architecture-parameters $\alpha$. We will hereafter refer to $\alpha$'s as NAS parameters. Each edge in the DARTS search space can assume any of the operations from the set of choices ($\mathcal{O}$), resulting in a mixed output defined by

$$\bar{m}(x^i) = \sum_{o \in \mathcal{O}} \frac{e^{\alpha_o}}{\sum_{o'} e^{\alpha_{o'}}} o(x^i), \tag{1}$$

where $o(x^i)$ denotes the output of operation $o$ applied to feature map $x^i$. DARTS and DARTS-PT both attempt to solve the following expression via alternating gradient updates:

$$\min_{\alpha} \mathcal{L}_{\text{val}} \left( \arg\min_{\theta} \mathcal{L}_{\text{train}} (\theta, \alpha, \mathcal{U}), \alpha, \mathcal{V} \right). \tag{2}$$

In particular, the gradient with respect to $\alpha$ can be approximated via

$$\nabla_{\alpha} \mathcal{L}_{\text{val}} (\theta - \zeta \nabla_{\theta} \mathcal{L}_{\text{train}} (\theta, \alpha, \mathcal{U}), \alpha, \mathcal{V}), \tag{3}$$

which can then be optimized using alternating gradient descent updates, according to a hyperparameter $\zeta$ (learning rate).

Once the supernetwork finishes training via gradient descent, the continuous NAS parameters $\alpha$ must be *discretized*. In the original DARTS algorithm, this is achieved by taking the largest $\alpha_o$ on each edge. However, Wang et al. (2021) showed that this approach may not perform well. Instead, at each edge, DARTS-PT directly evaluates the strength of each operation by its contribution to the supernetwork's performance, using a perturbation-based scoring technique (Wang et al., 2021). Now that we have given an overview of one-shot methods, in the next section we give an overview of adaptive subset selection methods, before combining them in Sec. 3.3

### 3.2 Adaptive Subset Selection Preliminaries

GLISTER. (GeneraLIzation based data Subset selecTion for Efficient and Robust learning), is a subset selection algorithm that adaptively selects a subset of the training data that maximizes the log-likelihood on a held-out validation set. This problem is formulated as a mixed discrete-continuous bi-level optimization problem. GLISTER approximately solves the following expression by first approximating the bi-level optimization expression using a single gradient step, and then using a greedy data subset selection procedure (Killamsetty et al., 2020).

$$\min_{S \subseteq \mathcal{U}, |S| \leq k} \mathcal{L}_{\text{val}} \left( \arg\min_{\theta} \mathcal{L}_{\text{train}} (\theta, S), \mathcal{V} \right). \tag{4}$$

In particular, the validation loss is approximated as follows:

$$\mathcal{L}_{\text{val}}\left(\arg\min_{\theta}\mathcal{L}_{\text{train}}\left(\theta,S\right),\mathcal{V}\right) \approx \mathcal{L}_{\text{val}}\left(\theta - \zeta\nabla_{\theta}\mathcal{L}_{\text{train}}\left(\theta,S\right),\mathcal{V}\right). \tag{5}$$

Thereafter, a simple greedy dataset subset selection procedure is employed to find the subset $S$ which approximately minimizes the validation loss (Killamsetty et al., 2020). Now that we have gone over the preliminaries of one-shot NAS and adaptive subset selection, we combine them in the next section. Our main approach makes use of GLISTER, but we also consider GRAD-MATCH, another adaptive subset selection approach, as an ablation study in Appendix A.4.4.

### 3.3 Our Framework

ADAPTIVE-DPT. There exist multiple possibilities for applying adaptive subset selection to one-shot NAS. We present GLISTER here (the best-performing method in our experiments) and GRAD-MATCH in Appendix A.4.4. Specifically, we now present a formulation that organically combines Equation (2) and Equation (4) into a single mixed discrete and continuous bi-level optimization problem. The inner optimization is the minimization (over model weights $\theta$) of training loss during architecture training, on a subset of the training data of size $k$. In the outer optimization, we minimize the validation loss by simultaneously optimizing over the NAS parameters $\alpha$ as well as over the subset of the training data $S$. This optimization problem is aimed at efficiently determining the best (or, at least, an effective) neural architecture:

$$\arg\min_{S\subseteq\mathcal{U},|S|\leq k,\alpha}\mathcal{L}_{\text{val}}\left(\arg\min_{\theta}\mathcal{L}_{\text{train}}\left(\theta,\alpha,S\right),\alpha,\mathcal{V}\right). \tag{6}$$

Evaluating this expression is computationally prohibitive because of the expensive inner optimization problem. Instead, we iteratively perform a joint optimization of the weights $\theta$ from the inner optimization as well as the training subset $S$ and NAS parameters $\alpha$ from the outer optimization. In order to iteratively update the training subset and architecture, we compute meta-approximations of the inner optimization. As for the architecture, we compute

$$\nabla_{\alpha}\mathcal{L}_{\text{val}}\left(\arg\min_{\theta}\mathcal{L}_{\text{train}}\left(\theta,\alpha,S\right),\alpha,\mathcal{V}\right) \approx \nabla_{\alpha}\mathcal{L}_{\text{val}}\left(\theta - \zeta\nabla_{\theta}\mathcal{L}_{\text{train}}\left(\theta,\alpha,S\right),\alpha,\mathcal{V}\right). \tag{7}$$

For the subset selection optimization problem, following Killamsetty et al. (2020), we run a greedy algorithm on a similar approximation to the inner optimization:

$$\mathcal{L}_{\text{val}}\left(\arg\min_{\theta}\mathcal{L}_{\text{train}}\left(\theta,\alpha,S\right),\alpha,\mathcal{V}\right) \approx \mathcal{L}_{\text{val}}\left(\theta - \zeta\nabla_{\theta}\mathcal{L}_{\text{train}}\left(\theta,\alpha,S\right),\alpha,\mathcal{V}\right). \tag{8}$$

The above equations form the basis of our method: we iteratively update the outer parameters (architecture and subset) using the approximations above, and the inner parameters (weights) using standard gradient descent; also see Algorithm 1 for the details of our approach. Following prior work (Killamsetty et al., 2020; Liu et al., 2019), we only update the architecture and subset every $t_1$ and $t_2$ steps, respectively, for efficiency ($t_1 << t_2$).

## 4 Experiments

In this section, we describe our experimental setup and results.

### 4.1 Experimental setup and details

**Search spaces**. We perform experiments on NAS-Bench-201 with CIFAR-10, CIFAR-100, and ImageNet16-120, DARTS with CIFAR-10, and DARTS-S4 with CIFAR-10. See Appendix A.3 for more information on the search spaces.

Table 1: Performance of one-shot NAS algorithms on NAS-Bench-201 CIFAR-10.

| Performance on NAS-Bench-201 CIFAR-10 | | | |
|---|---|---|---|
| Algorithm | Test accuracy | GPU hours | % Data used |
| DARTS-PT | 88.21 (88.11) | 7.50 | 100 |
| DARTS-PT-ENTROPY | 86.31 ± 4.66 | **0.62** | 10 |
| DARTS-PT-RAND | 86.94 ± 3.58 | **0.62** | 10 |
| DARTS-PT-FL | 89.27 ± 1.09 | 1.60 | 10 |
| ADAPTIVE-DPT | **92.22** ± 1.76 | 0.83 | 10 |

Table 2: Performance of one-shot NAS algorithms on NAS-Bench-201 CIFAR-100.

| Performance on NAS-Bench-201 CIFAR-100 | | | |
|---|---|---|---|
| Algorithm | Test accuracy | GPU hours | %Data used |
| DARTS-PT | 61.650 | 8.00 | 100 |
| DARTS-PT-ENTROPY | 56.79 ± 7.63 | **0.58** | 10 |
| DARTS-PT-RAND | 56.95 ± 4.54 | **0.58** | 10 |
| DARTS-PT-FL | 64.28 ± 3.10 | 0.67 | 10 |
| ADAPTIVE-DPT | 64.27 ± 3.37 | 0.87 | 10 |
| ADAPTIVE-DPT(F) | **64.83** ± 4.70 | 0.67 | 10 |

Table 3: Performance of one-shot NAS algorithms on NAS-Bench-201 ImageNet16-120.

| Performance on NAS-Bench-201 Imagenet16-120 | | | |
|---|---|---|---|
| Algorithm | Test accuracy | GPU hours | %Data used |
| DARTS-PT | 35.00 | 33.50 | 100 |
| DARTS-PT-ENTROPY | 26.52 ± 3.73 | **1.58** | 10 |
| DARTS-PT-RAND | 27.04 ± 5.53 | **1.58** | 10 |
| DARTS-PT-FL | 29.30 ± 5.35 | 1.90 | 10 |
| ADAPTIVE-DPT | **36.10** ± 6.96 | 2.60 | 10 |

Table 4: Performance of one-shot NAS algorithms on S4 search space CIFAR-10.

| Performance on S4 CIFAR-10 | | | |
|---|---|---|---|
| Algorithm | Test accuracy | GPU hours | %Data used |
| DARTS-PT | 97.31 (97.36) | 8.38 | 100 |
| DARTS-PT-ENTROPY | **97.45** ± 0.10 | **0.86** | 10 |
| DARTS-PT-RAND | 97.40 ± 0.06 | **0.86** | 10 |
| DARTS-PT-FL | 97.34 ± 0.13 | 1.08 | 10 |
| ADAPTIVE-DPT | 97.30 ± 0.12 | 1.08 | 10 |

**Algorithm 1** Pseudocode for Adaptive-Dpt. *GreedyDSS* refers to the set of greedy algorithms and approximations that solves Equation (6); see Appendix A.4.1 for its full details.

---

1: **Require**: Training data $\mathcal{U}$, Validation data $\mathcal{V}$, Initial subset $S^{(0)}$ of size $k$, Initial parameters $\theta^{(0)}$ and $\alpha^{(0)}$, steps $T_1$, $T_2$, and $T$.
2: for all steps $t$ in $T$ do
3:    if $t$ mod $T_1 == 0$:
4:      $S^{(t)} = \text{GreedyDSS}(\mathcal{U}, \mathcal{V}, \theta^{(t-1)}, \alpha^{(t-1)})$
5:    else:
6:      $S^{(t)} = S^{(t-1)}$
7:    if $t$ mod $T_2 == 0$:
8:      Perform one step of SGD to update $\alpha^{(t)}$ using $V$
9:    else:
10:     $\alpha^{(t)} = \alpha^{(t-1)}$
11:    Perform one step of SGD to update $\theta^{(t)}$ using $S^{(t)}$ and $\alpha^{(t)}$.
12: Discretize the supernet, based on NAS parameters $\alpha^{(T)}$ obtained using $S^{(T)}$, to return final architecture
13: Train $\alpha$ using SGD with the full training set $\mathcal{U}$
14: **Return**: Final architecture (discretized $\alpha^{(T)}$)

---

Table 5: Performance of one-shot NAS algorithms on DARTS search space CIFAR-10.

| Performance on DARTS CIFAR-10 | | | |
|---|---|---|---|
| Algorithm | Test accuracy | GPU hours | %Data used |
| Darts-pt | 97.17 (97.39) | 20.59 | 100 |
| Darts-pt-entropy | 96.68 ± 0.26 | 3.40 | 10 |
| Darts-pt-rand | **97.01** ± 0.32 | **2.35** | 10 |
| Darts-pt-fl | 96.91 ± 0.15 | 4.00 | 10 |
| Adaptive-Dpt | 96.73 ± 0.29 | 2.75 | 10 |
| Adaptive-Dpt | 96.97 ± 0.24 | 4.50 | 20 |
| Adaptive-Dpt | **97.30** ± 0.05 | 8.82 | 30 |

**Baselines and approaches tested.** We perform experiments with DARTS-PT, Adaptive-Dpt, and three other (non-adaptive) data subset selection methods applied to DARTS-PT (Wang et al., 2021). We describe the details of each approach below.

- Darts-pt: We use the original implementation of DARTS-PT, as described in Sec. 3.

- Darts-pt-rand: This is similar to Darts-pt, but the supernetwork is trained and discretized using a random subset of the training data.

- Darts-pt-fl: While similar to Darts-pt, the supernetwork is trained and discretized using a subset of the training data, selected using facility location. The facility location function attempts to find a representative subset of items. The function is similar to k-medoid clustering: for each data point $i$ in the ground set $V$, we compute the representative from subset $X$ which is closest to $i$ and add these similarities for all data points.

$$f(X) = \sum_{i \in \mathcal{V}} \max_{j \in \mathcal{X}} s_{ij} \tag{9}$$

The facility location algorithm was implemented using the naive greedy algorithm and ran on each class separately, using a dense Euclidean metric. For this, we employed the submodlib library (Kaushal et al., 2022).

- Darts-pt-entropy (Na et al., 2021): again, this bears similarity to Darts-pt but using a non-adaptive subset selection procedure. Specifically, the subset of the training data is selected using a combination of high and low-entropy datapoints. Unlike the other existing zero cost subset selection methods for NAS, this approach is specifically tailored for NAS and accelerates neural architecture search using proxy data. The entropy of a datapoint is calculated by training a base neural architecture from the search space, and determining whether the output probability is low or high. This approach was adopted by Na et al. (2021), to speed up DARTS.

- Adaptive-Dpt: This is one of our key approaches, as described in Algorithm 1.

**Experimental setup.** Following Wang et al. (2021), we use 50% of the full training dataset for supernet training and 50% for validation. We report the accuracy of the finally obtained architecture on the held-out test set. In our primary experiments, for each (adaptive or non-adaptive) subset selection method, we set the subset size to 10% of the training dataset. We run the same experimental procedure for each method: select a size-10% subset of the full training dataset, train and discretize the supernet on the subset, and train the final architecture using the full training dataset. For Darts-pt, we run the same procedure using the full training dataset at each step. We otherwise use the exact same training pipeline as in Wang et al. (2021), i.e, batch size of 64, learning rate of 0.025, momentum of 0.9, and cosine annealing. We run all experiments on an NVIDIA Tesla V100 GPU. We run each algorithm with 5 random seeds, reporting the mean and standard deviation of each method, with the exception of Darts-pt; due to its extreme runtime and availability of existing results, we perform the experiment once and verify that the result is nearly identical to published results (Wang et al., 2021). We also report the time it takes to output the final architecture.

### 4.2 Adaptive-Dpt results and discussion

In this section, we give results for Adaptive-Dpt across the search spaces and baselines described above. We start with the core results, then we present ablations, and then we describe the key takeaways.

In Tab. 1, Tab. 2, and Tab. 3, we report the results on NAS-Bench-201. On CIFAR-10 and ImageNet16-120, Adaptive-Dpt yields higher accuracy than all other algorithms tested, and on CIFAR-100, Adaptive-Dpt is essentially tied with Darts-pt-fl for the highest accuracy. Furthermore, all NAS algorithms that use subset selection have significantly decreased runtime – Adaptive-Dpt sees a factor of 9 speedup compared to Darts-pt. Note that Darts-pt-fl takes more time when the number of examples per class in the dataset is higher, so it sees comparatively higher runtimes on CIFAR-10.

Next, in Tab. 4 and Tab. 5, we report the results on S4 CIFAR-10 and DARTS CIFAR-10. Once again, the runtime of Adaptive-Dpt is significantly faster than Darts-pt – a factor of 7 speedup. On these search spaces, the performances of the subset-based methods are more similar when compared to NAS-Bench-201, and on the DARTS search space, Adaptive-Dpt does not outperform Darts-pt. A possible explanation is that S4 and DARTS are significantly larger search spaces than NAS-Bench-201 and require more training data to distinguish between architectures. To test this, we included an additional experiment in Tab. 5, giving Adaptive-Dpt 20% and 30% training data instead of 10%. We find that the accuracy significantly increases, moving within one standard deviation of the accuracy of Darts-pt for 20% and slightly better than Darts-pt for 30%.

Additionally, since Jain et al. (2023) ran the same experiment as us on the DARTS CIFAR-10, we compare our results in Tab. 5 to their Table 4. Adaptive-Dpt achieves 96.73 ± 0.29%, 96.97 ± 0.24%, and 97.30 ± 0.05% on 10%, 20%, and 30% data, compared to SubSelNet's 97.18%,

97.24%, and 97.32% across 10%, 20%, and 40% data. Note a caveat: since we are comparing to results in their paper, we only accuracy with respect to subset size, and although the experimental setup appears to be similar, we cannot guarantee it is a fully apples-to-apples comparison. That said, Adaptive-Dpt performs within the standard deviation of the accuracy of SubSelNet, and achieves near-identical performance with 30% data vs. their 40% data. Overall, Adaptive-Dpt achieves near-top performance across all search spaces, while also achieving no less than a seven-fold increase in runtime compared to Darts-pt.

**Ablation study**. Next, we briefly describe three ablation studies: using Grad-Match rather than Glister, computing the accuracy of Adaptive-Dpt as a function of subset sizes, and performing the projection step with/without the full training dataset. We present these ablation studies in Appendix A.1, and we summarize the results here.

In Appendix A.4.4, we report the results of DARTS-PT with adaptive subset selection using Grad-Match (Killamsetty et al., 2021) on NAS-Bench-201 with datasets CIFAR-10 and CIFAR-100. Although DARTS-PT with Grad-Match could not outperform Adaptive-Dpt, it outperformed most non-adaptive subset selection methods.

In Appendix A.1, we test the performance of Adaptive-Dpt using different percentages of the training data, ranging from 1% to 50%, on NAS-Bench-201 CIFAR10. Interestingly, we see a U-shape: the highest accuracy with Adaptive-Dpt is at 20%, achieving accuracy *higher* than the standard setting of 100% data. Since the supernetwork is an over-parameterized model of weights and architecture parameters, and Adaptive-Dpt regularly updates the training subset to maximize validation accuracy, Adaptive-Dpt may help prevent the supernetwork from overfitting. Finally, we compare Adaptive-Dpt with and without using the full training dataset for the projection (perturbation) step of DARTS-PT, where the default for Adaptive-Dpt is to use a subset of the data. We find that there is high increase in the runtime, and accuracy is moderately improved on DARTS CIFAR10, and remains similar for S4. We present the full details in Appendix A.1.

**Key takeaways**. Overall, we find that across many settings, Adaptive-Dpt improves the runtime by roughly an order of magnitude, without sacrificing accuracy. While using 10% of the training data is a strong default, the user may decide their desired tradeoff between performance and accuracy, and choose the subset size in the supernetwork training accordingly.

### 4.3 Additional extensions

In this section, we discuss additional extensions beyond our core results and ablations above. Specifically, we consider experiments with additional NAS algorithms (DEHB and BOHB), and additional regularisation with other subset selection methods.

**Adaptive-DEHB and Adaptive-BOHB**. To show the generality of our approach, we combine DEHB (Awad et al., 2021) and BOHB (Falkner et al., 2018b) with adaptive subset selection (Glister). These are (efficient) iterative algorithms, meaning that we can simply run DEHB and BOHB as normally, while iteratively selecting the best subset using Equation (4) and Equation (5), rather than the more involved optimization procedure that we used for one-shot methods. See Appendix A.5 for more details. We use the same search space as Awad et al. (2021): a configuration sampled from a parameter space (with parameters such as kernel size, channel size, and stride) is used to instantiate a base CNN architecture. On this architecture, we trained DEHB on the MNIST dataset for 100 epochs with and without subset selection. Across 5 seeds, DEHB trained without adaptive subset selection took **0.91** hours and gave **0.96 ± 0.03** accuracy whereas Adaptive-DEHB using 20% data and selecting subset at every 10 epochs took **0.85** hours and yielded **0.99 ± 0.00** accuracy. For Fashion-MNIST (Xiao et al., 2017), across 3 seeds, DEHB took **1.43** hours and gave **0.85 ± 0.03**, while Adaptive-DEHB using 20% data took **0.98** hours and yielded **0.94 ± 0.03**.

For BOHB on the MNIST dataset, one set of experiments was done with this complete data and another with a subset of these selected by Glister every 10 epochs. BOHB without adaptive

subset selection gave an accuracy of **0.99 ± 0.00** and took **2.43** hours on MNIST dataset whereas ADAPTIVE-BOHB gave an accuracy of **0.98 ± 0.00** and took **1.16** hours with 20% data.

**Regularisation with fixed subset selection functions**. In Sec. 4.2, we showed that ADAPTIVE-DPT can sometimes outperform DARTS-PT, due to a (likely) regularization effect. Now, we push this further by applying additional fixed subset selection. Specifically, we add a fixed subset selection function, $R(S)$ to our original objective, along with a $\lambda$ tradeoff parameter:

$$\min_{S \subseteq \mathcal{U}, |S| \leq k} \mathcal{L}_{\text{val}} \left( \arg \min_{\theta} \mathcal{L}_{\text{train}} \left( \theta, S \right), \mathcal{V} \right) + \lambda \cdot R(S) \tag{10}$$

Then, we run the method the same way as before (Algorithm 1). We considered two regularization functions. First is a random function, ADAPTIVE-DPT(R) (selecting 10% random data) and second is supervised facility location, ADAPTIVE-DPT(F) which is NN-submodular function on the training set features. We tested on CIFAR-100 and found that adaptive subset selection using regularised (supervised facility location) GLISTER outperforms all models on CIFAR-100 with a tradeoff on time. For ADAPTIVE-DPT(R) and ADAPTIVE-DPT(F), on CIFAR-100, we used 10% fixed subset selection and 10% adaptive subset selection. ADAPTIVE-DPT(R) achieved an accuracy of **60.40 ± 2.60** in **0.88** hours, while ADAPTIVE-DPT(F) achieved an accuracy of **65.41 ± 2.57** in **1.1** hours. This approach shows that we can use non-adaptive subset selection algorithms as regularization functions for adaptive subset selection algorithms to achieve better test accuracy.

# 5 Conclusions, Future work, and Impact

In this work, we used a connection between one-shot NAS algorithms and adaptive subset selection to devise an algorithm that makes use of state-of-the-art techniques from both areas. Specifically, we build on state-of-the-art approaches to one-shot NAS and adaptive subset selection and pose a bi-level optimization problem on the training and validation sets. This leads us to the formulation of a combined approach, ADAPTIVE-DPT, as a mixed discrete and continuous bi-level optimization problem. We empirically demonstrated that the resulting algorithm is able to train on an (adaptive) dataset that is 10% of the size of the full training set, without sacrificing accuracy, resulting in an order of magnitude decrease in runtime. We also show how this method can be extended to hyperparameter optimization algorithms, in general, using ADAPTIVE-DEHB and ADAPTIVE-BOHB. We also release a codebase consisting of four different subset selection techniques.

**Future work**. An interesting direction for future work is to use an adaptive subset of the data even when training the final architecture, which may lead to even faster runtime, perhaps at a small cost to performance. Another interesting direction for future work is to apply adaptive subset selection to other NAS algorithms such as regularized evolution (Real et al., 2019) or additional supernet-based algorithms (see, e.g., (Heuillet et al., 2023)). Another interesting direction is to apply ADAPTIVE-DPT to a transformer-based search space such as in Section 7.4 in White et al. (2023).

**Broader impact**. Our work combines techniques from two different areas: adaptive subset selection for machine learning, and neural architecture search. The goal of our work is to make it easier and quicker to develop high-performing architectures on new datasets. Our work also helps to unify two sub-fields of machine learning that had thus far been disjoint. There may be even more opportunity to use tools from one sub-field to make progress in the other sub-field, and our work is the first step at bridging these subfields.

Since the end product of our work is a NAS algorithm, it is not itself meant for one application but can be used in any end-application. Our hope is that future AI models discovered by our work will have a net positive impact, due to the push for the AI community to be more conscious about the societal impact of its work (Hecht et al., 2018).

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

Table 6: Performance of Adaptive-Dpt on the DARTS search space and the S4 search space.

| Performance on CIFAR-10 | | | |
|---|---|---|---|
| Search Space | Test accuracy | GPU hours | % Data used |
| DARTS | $97.06 \pm 1.09$ | 2.65 | 10 |
| S4 | $97.28 \pm 0.03$ | 1.18 | 10 |

Table 7: Performance and runtime of Adaptive-Dpt varies as the percentage of training data increases, on NAS-Bench-201 search space CIFAR-10. The supernetwork training and projection step are given a percentage of the full training dataset.

| Performance on NAS-Bench-201 CIFAR-10 | | |
|---|---|---|
| Test accuracy | GPU hours | %Data used |
| $57.91 \pm 43.77$ | 0.30 | 1 |
| $89.21 \pm 3.12$ | 0.35 | 2 |
| $91.95 \pm 1.56$ | 0.58 | 5 |
| $92.22 \pm 1.76$ | 0.83 | 10 |
| $\mathbf{92.24 \pm 1.65}$ | 1.58 | 20 |
| $88.23 \pm 0.06$ | 3.63 | 50 |

## A  Additional Results and Analyses

In this section, we give additional results and analyses to supplement Sec. 4.

### A.1  Ablation study

First, we describe the ablation studies (which were summarized in Sec. 4.2).

To explore the effect of the percentage of data used, in Figure 2 (left), we run Adaptive-Dpt with different percentages of the training data, ranging from 1% to 50%. In Figure 2 (right), we run the same experiment using the full training data in the projection step of Adaptive-Dpt. Interestingly, we see a definitive U-shape in the first experiment: the highest accuracy with Adaptive-Dpt is at 20%, achieving accuracy *higher* than the standard setting of 100% data. Since the supernetwork is an over-parameterized model of weights and architecture parameters, and Adaptive-Dpt regularly updates the training subset to maximize validation accuracy, Adaptive-Dpt may help prevent the supernetwork from overfitting. Furthermore, in the second experiment, we see that relatively, the accuracies are much more consistent when varying the percentage of the training set used, when the projection step is allowed to use the full training dataset. Therefore, keeping the full training dataset for the projection step leads to higher and more consistent performance, at the expense of more GPU-hours.

Overall, based on the ablation studies in Figure 2, the user may decide their desired tradeoff between performance and accuracy, and choose the subset size in the supernetwork training accordingly. For example, with a budget of 1 GPU hour, the best approach is to use a 10% subset of the training data for the supernet training and projection, but with a budget of 2.5 GPU hours, the best approach is to use a 10% subset of the training data for the supernet and the full training data for the projection.

In Tab. 6, we give the results of Adaptive-Dpt with the full data for the DARTS-PT projection step for search spaces DARTS and S4. Although we were able to get better accuracy (when compared 10% data on projection step) on DARTS space, the accuracy stayed similar for S4.

In Tab. 7 and Tab. 8, we give the tables to match the plots in Figure 2.

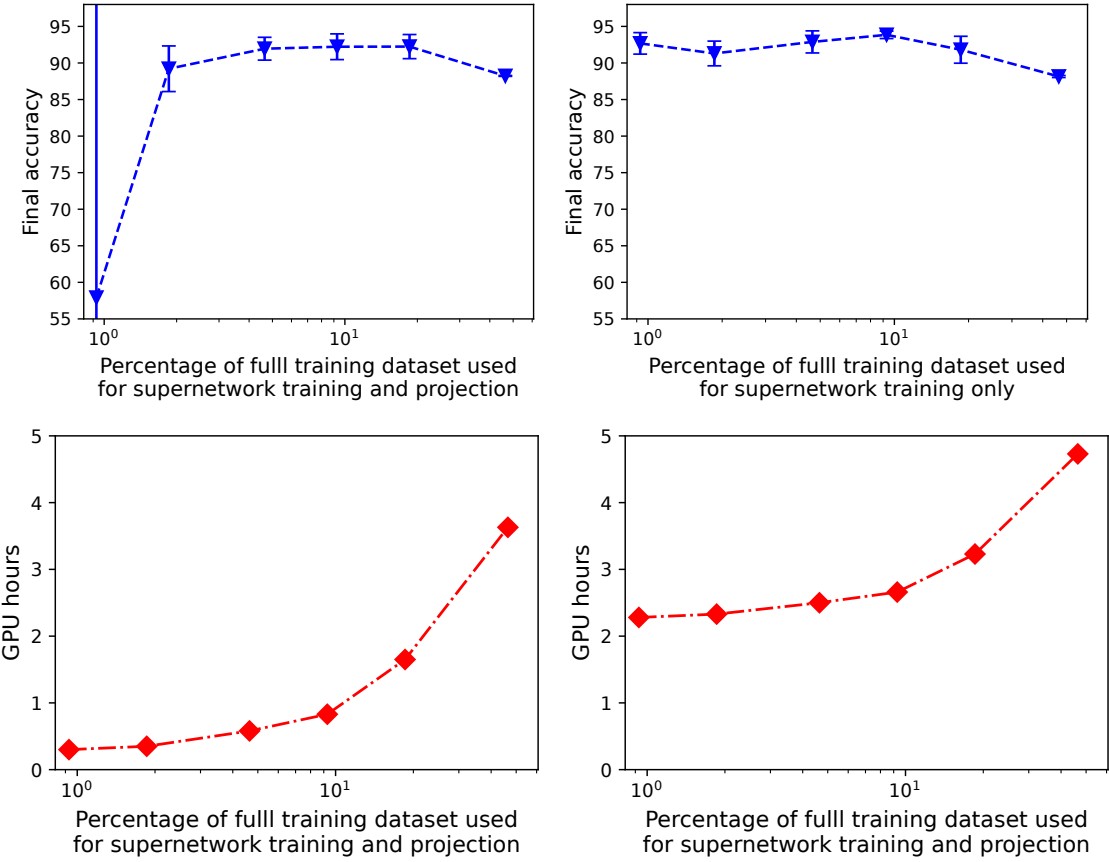

Figure 2: Performance and runtime of ADAPTIVE-DPT varies as the percentage of training data increases. (Left) The supernetwork training and projection step are given a percentage of the full training dataset. (Right) The supernetwork training is given a percentage, while the projection step is given the full training dataset.

In Tab. 9, we give the results of ADAPTIVE-DPT using the full data for the DARTS-PT projection (perturbation) step for search space NAS-Bench-201 for datasets CIFAR-100 and Imagenet16-120. Although we were able to get better accuracy (when compared to ADAPTIVE-DPT without full data for the perturbation step), the time taken was significantly more.

In Tab. 10, we give a summary of the improvements of ADAPTIVE-DPT when compared to DARTS-PT.

## A.2 Details from Sec. 4

In this section, we give more details for the experiments conducted in Sec. 4.

### A.2.1 Experiments on NAS-Bench-201.
We used the original code from DARTS-PT (Wang et al., 2021) and GLISTER (Killamsetty et al., 2020). The DARTS-PT code consists of two parts. The supernet training and a perturbation based projection part to discretize $\alpha$. The Supernet training is run for 100 epochs and at each 10 epoch interval, we select a new subset of data by passing the model and architecture parameters. At every epoch, we use 10% of the original dataset. We use a batchsize of 64, learning rate of 0.025, momentum of 0.9, and cosine annealing. We use 50% of data for training and 50% for validation, as in the DARTS-PT paper (Wang et al., 2021). The last 10% data subset is saved and used for the perturbation based projection part of DARTS-PT. We run the projection part

Table 8: Performance and runtime of ADAPTIVE-DPT varies as the percentage of training data increases, on NAS-Bench-201 search space CIFAR-10. The supernetwork training is given a percentage, while the projection step is given the full training dataset.

| Performance on NAS-Bench-201 CIFAR-10 | | |
|---|---|---|
| Test accuracy | GPU hours | %Data used |
| 92.68 ± 1.47 | 2.28 | 1 |
| 91.30 ± 1.69 | 2.33 | 2 |
| 92.88 ± 1.51 | 2.50 | 5 |
| **93.85** ± 0.51 | 2.66 | 10 |
| 91.81 ± 1.84 | 3.23 | 20 |
| 88.15 ± 0.14 | 4.73 | 50 |

Table 9: Performance of ADAPTIVE-DPT on NAS-Bench-201.

| Performance on NAS-Bench-201 | | | |
|---|---|---|---|
| Dataset | Test accuracy | GPU hours | % Data used |
| CIFAR-100 | 65.85 ± 4.17 | 2.43 | 10 |
| Imagenet16-120 | 37.43 ± 2.12 | 8.78 | 10 |

Table 10: Summary of Improvements over DARTS-PT by ADAPTIVE-DPT, across all search spaces.

| Summary of Improvements | | | | |
|---|---|---|---|---|
| Search Space | Dataset | Accuracy | Time reduced | % Data Used |
| NAS-Bench-201 | CIFAR-10 | +5.07 | 8.62 times | 10 |
| NAS-Bench-201 | CIFAR-100 | +2.63 | 9.20 times | 10 |
| NAS-Bench-201 | Imagenet-16-120 | +1.10 | 12.80 times | 10 |
| S4 | CIFAR-10 | -0.01 | 7.76 times | 10 |
| DARTS | CIFAR-10 | +0.13 | 2.33 times | 30 |

for 25 epochs. For subset selection, we used the same code of GLISTER with selection algorithm run on each class separately.

For Darts-pt-fl, we used the implementation of Facility Location as present in `submodlib`. This subset selection algorithm was used in the dense Euclidean setting. The algorithm is used separately for each class so as to keep the representation across classes the same as original. It was optimised using the 'NaiveGreedy' algorithm, For the experiments, 10% data was used.

For Darts-pt-rand and Darts-pt-entropy, we combined Darts-pt with proxy data using two methods of subset selection techniques for dataset, one a random subset selection and other an entropy based subset selection technique (Na et al., 2021). For random subset data was chosen randomly from the dataset. For the entropy based selection, we used the entropy files for CIFAR-10 and CIFAR-100 from Na et al. (2021) which was obtained by training a baseline network of ResNet20 and ResNet56 respectively. For ImageNet16-120, we trained a ResNet-50 model from the PyTorch model zoo.

For S4 and the DARTS search space, we used the same configuration as for NAS-Bench-201. Since S4 and DARTS are non-tabular, we used a separate evaluation code for computing the performance of the selected architecture. We used the same evaluation code given in Darts-pt. The code uses a batch size of 96, learning rate of 0.025, momentum of 0.9 and weight decay of 0.025. The architecture is trained for 600 epochs.

## A.3 Additional Details of the Search Spaces

**A.3.1 NAS-Bench-201.** In NAS-Bench-201 (Dong and Yang, 2020), the search space is based on cell-based architectures where each cell is a DAG. Here each node is a feature map and each edge is an operation. The search space for NAS-Bench-201 is defined by 4 nodes and 5 operations making 15625 different cell candidates.

NAS-Bench-201 gives performance of every candidate architecture on three different datasets (CIFAR-10, CIFAR-100, Imagenet-16-120). This makes NAS-Bench-201 a fair benchmark for the comparison of different NAS algorithms. The five representative operations chosen for NAS-Bench-201 are: (1) zeroize (dropping the associated edge) (2)skip connection (3) 1-by-1 convolution (4) 3x3 convolution (5) 3x3 average pooling layer. Each convolution operation is a sequence of ReLU, convolution and batch normalization. The input of each node includes the sum of all the feature maps transformed using the respective edge operations. Each candidate architecture is trained using Nestorov momentum SGD using cross entropy loss for 200 epochs.

**A.3.2 DARTS-CNN search space.** The search space is represented using cell based architectures (Liu et al., 2019). Each cell is a DAG with feature maps as nodes and edges as operations. The operations included are 3x3 and 5x5 separable convolutions, 3x3 and 5x5 dilated separable convolutions, 3x3 max pooling, 3x3 average pooling, identity and zero. Each cell consists of 7 nodes where output node is depth-wise concatenation of all the intermediate nodes.

**A.3.3 DARTS S4.** S1-S4 are four different search spaces proposed by Zela et al. (2020). These search spaces were proposed to demonstrate the failure of standard DARTS. The same micro-architecture as the original DARTS paper with normal and reduction cells is used but only a subset of operators are allowed for the search spaces. The representative set of operations for S4 is {3x3 SepConv, Noise}. SepConv is chosen since it is one of the most common operation in the discovered cells reported by Liu et al. (2019). Noise operation plugs in the noise values $\epsilon \sim N(0, 1)$ for every value from the input feature map.

## A.4 Additional Details of the Algorithms Implemented

In this section, we give more details for GLISTER and the baselines used in Sec. 4.

**A.4.1 Details of GLISTER.** The optimization that we are trying to solve for GLISTER, equation (4), can be written as

$$S^{t+1} = \underset{S \subseteq U, |S| \leq k}{\arg\min} \ G_{\theta^t}(S) \tag{11}$$

where $G_{\theta^t}(S)$ is Equation (5). Since equation (11) is an instance of submodular optimization (as proven in Theorem 1 of Killamsetty et al. (2020)), it can be regularized using another function such as supervised facility location. The regularized objective can be written as

$$S^{t+1} = \underset{S \subseteq U, |S| \leq k}{\arg\min} \ G_{\theta^t}(S) + \lambda R(S) \tag{12}$$

where $\lambda$ is a trade-off parameter. GreedyDSS refers to the set of greedy algorithms and approximations that solves Equation (12). Greedy Taylor Approximation algorithm (GreedyTaylorApprox(U, V, $\theta^0$, $\eta$, k, r, $\lambda$, R), described as Algorithm 2 in Killamsetty et al. (2020)) is used as GreedyDSS in our work. Here, $U$ and $V$ are the training and validation set respectively. $\theta^t$ is the current set of parameters, $\eta$ is the learning rate, $k$ is the budget, parameter $r$ governs the number of times we perform the Taylor series approximation, and $\lambda$ is the regularization constant.

**A.4.2 Details of facility location.** Intuitively, facility location, attempts to model representation of the datapoints. If $s_{ij}$ is the similarity between datapoints $i$ and $j$, define $f(X)$ such that

$$f(X) = \sum_{i \in V} \max_{j \in X} s_{ij} \tag{13}$$

where $V$ is the ground set. If the ground set of items are assumed clustered, an alternative clustered implementation of facility location is computed over the clusters as

$$f(X) = \sum_{l \in 1..K} \sum_{i \in C_l} \max_{j \in X \cap C_l} s_{ij} \tag{14}$$

**A.4.3 Details of DARTS-PT-ENTROPY.** DARTS-PT-ENTROPY is the implementation of Na et al. (2021) where the cost of NAS is reduced by selecting a representative set of the original training data. The entropy of a datapoint is defined as

$$\text{Entropy}(x : M) = -\sum_{\tilde{y}} p(\tilde{y}|x, M) \log p(\tilde{y}|x, M) \tag{15}$$

where $\tilde{y} = M(x)$ is the predictive distribution of $x$ w.r.t. a pre-trained baseline model $M$.

The selection method selects datapoints from both the high entropy and low entropy regions.

If $h_x$ is a bin of the data point $x$ on data entropy histogram $H$, $|h_x|$ is the height of $h_x$ (number of examples in $h_x$), three probabilities are defined as

$$P_{\{1,2,3\}}(x; H) = \text{norm}(W_{\{1,2,3\}}(h_x; H)/|h_x|) \tag{16}$$

where $\text{norm}()$ is a normalizer such that the probability terms add to 1. $W_{\{1,2,3\}}$ are selected such that the tail end entropy data are likely to be selected over the middle entropy data points.

In Na et al. (2021), $P_1(x)$ was the highest performer. We have used $P_1(x)$ in our experiments.

**A.4.4 Details of GRAD-MATCH.**

Table 11: Performance of DARTS-PT + GRAD-MATCH on NAS-Bench-201

| Performance on NAS-Bench-201 CIFAR-10 | | | |
|---|---|---|---|
| Dataset | Test accuracy | GPU hours | % Data used |
| CIFAR-10 | 88.83 ± 1.09 | 0.87 | 10 |
| CIFAR-100 | 63.70 ± 3.98 | 0.87 | 10 |

**GRAD-MATCH.** GRAD-MATCH, Gradient Matching based Data Subset Selection for Efficient Deep learning Model Training is proposed in Killamsetty et al. (2021). GRAD-MATCH selects a subset that best approximates either the full training dataset (or) a held-out validation dataset. This is achieved by selecting a coreset whose gradient matches the average loss gradient over the training dataset or the validation dataset respectively. The objective is modelled as a discrete subset selection problem that is combinatorially hard to solve and in response, they propose Orthogonal Matching Pursuit based greedy algorithm to pick the subset.

In Killamsetty et al. (2021), it is proved that the convergence bounds of adaptive subset selection algorithms critically depends on an additive error term that depends on how well the subset's gradients match the full training/validation gradient. The Error term E is defined as:

$$E_\lambda(\mathbf{w}, S) = \lambda\|\mathbf{w}\|^2 \tag{17}$$
$$+ \|\sum_{j \in S} \mathbf{w}_j \nabla_\theta \mathcal{L}_{\text{train}}(\theta, j) - \frac{1}{|\mathcal{U}|} \sum_{u \in \mathcal{U}} \nabla_\theta \mathcal{L}_{\text{train}}(\theta, u)\|$$

where $\mathbf{w} \in \mathbb{R}^k$ represents the weight coefficient attached to each point in the coreset($S$). First term of E is a squared $l_2$ loss regularizer over the weight vector with a regularizer coefficient $\lambda$ to prevent overfitting.

The objective function for the GRAD-MATCH version that selects a coreset to approximate training gradient is:

$$\underset{S \subseteq \mathcal{U}, |S| \leq k}{\text{argmin}} \ \underset{\mathbf{w}}{\min} E_\lambda(\mathbf{w}, S) \text{ where,} \tag{18}$$

$\mathbf{w} \in \mathbb{R}^k$ represents the weight coefficient attached to each point in the coreset. Essentially, the formulation selects a subset whose weighted sum of gradients match the average training gradient.

In Table 11, we give performance of DARTS-PT + GRAD-MATCH on NAS-Bench-201.

### A.5 ADAPTIVE-DEHB and ADAPTIVE-BOHB

**ADAPTIVE-DEHB.** Differential evolution hyperband (DEHB) (Awad et al., 2021) is a leading algorithm for mutli-fidelity optimization which has been applied to both hyperparameter optimization (HPO) and NAS. The approach combines differential evolution (Storn and Price, 1997), a population-based evolutionary algorithm, with hyperband (Li et al., 2018), a bandit-based multi-fidelity optimization routine which rules out poor hyperparameter settings before they are trained for too long. Unlike DARTS-based approaches, DEHB does not use a supernetwork – each architecture is trained separately. Therefore, to devise ADAPTIVE-DEHB, we incorporate adaptive subset selection simply by running GLISTER for each individual architecture trained throughout the algorithm.

**Adaptive-BOHB.** Bayesian Optimization and Hyperband (BOHB) (Falkner et al., 2018a) is a hyperparameter optimization method that combines benefits of Bayesian Optimization and bandit based methods (Li et al., 2018) such that it finds good solutions faster than Bayesian optimization and converges to the best solutions faster than Hyperband. We use adaptive subset selection along with BOHB to devise Adaptive-BOHB which yields accuracy almost similar to BOHB while reducing the runtime significantly.

### A.6 Supertickets and subset selection

The Lottery Ticket Hypothesis (Frankle and Carbin, 2018) and Supertickets (You et al., 2022) represent two complementary concepts in the realm of neural network pruning and efficiency. The Lottery Ticket Hypothesis posits that within a dense, over-parameterized neural network, there exist sparse subnetworks, or "winning lottery tickets," that can achieve similar performance with significantly fewer parameters. This hypothesis underscores the importance of identifying and training these critical subnetworks during the training process. Supertickets, on the other hand, build upon this idea by explicitly seeking out these winning subnetworks during training, rather than relying on the randomness of weight initialization. Supertickets are a strategic means to discover and harness the lottery ticket subnetworks, ensuring that the training process is not only more efficient but also yields models with improved resource utilization and competitive performance. In essence, both concepts share the common goal of reducing the computational and memory demands of deep learning models while preserving or even enhancing their predictive capabilities.

We combined adaptive subset selection to find supertickets (You et al., 2022) on MNIST dataset and found that the training time can be significantly reduced with better results.

The Pruning-Aware Critical (PrAC) (Zhang et al., 2021) subset selection approach is a key innovation aimed at identifying a subset of training examples that are essential for maintaining the performance of deep neural networks while significantly reducing the dataset size required for training. The PrAC approach focuses on discovering data instances that play a pivotal role in the learning process, aligning with the Lottery Ticket Hypothesis (Frankle & Carbin, 2018), which posits that sparse subnetworks within a neural architecture can be trained effectively. The goal is to create a data subset that facilitates the training of these sparse subnetworks, making it possible to achieve model efficiency with fewer training samples. To select PrAC subsets, the approach incorporates two crucial components: Critical Examples for Pruning (CEP) and Critical Examples for Training (CET). These subsets are carefully curated during the iterative magnitude pruning process and aim to strike a balance between data retention and model efficiency. Specifically, CET comprises examples that significantly affect the network's performance during training, while CEP focuses on data instances that are influential in the pruning process. By identifying and preserving these critical examples, PrAC ensures that the retained training data is not only conducive to network learning but also facilitates effective pruning.

We combined PrAC subset selection to find supertickets (You et al., 2022) on MNIST dataset and found that the training time can be significantly reduced with better results. In Tab. 12, we give performance of various subset selection algorithms on supertickets code (You et al., 2022).

Table 12: Performance of subset selection algorithms for finding Supertickets

| Performance on MNIST | | | |
|---|---|---|---|
| Subset Selection Algorithm | Loss | Top 1 error | % Top 5 error |
| Vanilla | 0.414 | 0.1072 | 0.004 |
| PrAC | 0.377 | 0.003 | 0.003 |
| GLISTER | 0.951 | 0.299 | 0.0185 |
| GRAD-MATCH | 0.9598 | 0.2937 | 0.0165 |

