# OpenReview forum: "Speeding up NAS with Adaptive Subset Selection"
_automl.cc/AutoML/2024/Conference — AutoML 2024_

### Official Review · Reviewer_BkQR · 2024-03-26

**Potential Impact On The Field Of Automl Rating:** 4
**Technical Quality And Correctness Rating:** 4
**Clarity Rating:** 2

**Summary Of Contributions:**

This paper introduces an algorithm (Adaptive-DPT) that combines one-shot NAS with an adaptive subset selection algorithm to reduce the overall runtime of the NAS process. Empirical evaluations on the NAS-Bench-201 benchmark show that Adaptive-DPT runs on subsets as small as 10% of the original training set, achieving an order of magnitude speed-ups compared to training on the full dataset while maintaining test set accuracy.

**Actions Required To Increase Overall Recommendation:**

Improve the clarity of the paper and provide more extensive results for the BOHB/DEHB cases.

**Clarity:**

The overall clarity of the paper is limited. While it provides a lot of content, it misses clear connections between the paragraphs and lacks in-depth explanations at the relevant parts in the main paper (e.g. the adaptive subset selection, which is one of the core points of Adaptive-DPT seems to be derived from equations 8/9 and 10/11 but without sufficient explanation). Instead, the reader is referred to the appendix.

While the empirical evaluations of the one-shot NAS methods are presented in much detail, the combination of the subset selection procedure with BOHB\DEHB is presented only briefly in lines 234-247, lacking clarity and details.

**Overall Review:**

Pros:
- The main idea of combining one-shot NAS and adaptive subset selection is sound.
- The proposed method achieves good results (large reduction in runtime, almost no loss of accuracy) on the NAS-Bench-201 benchmark against a leading one-shot NAS method.
- The proposed method is also benchmarked against several other subset selection algorithms (such as random sampling or facility location) and does fairly well.

Cons:
- In contrast to the one-shot evaluation, the empirical evaluations of the proposed method in combination with BOHB and DEHB are very limited, as it is just one dataset (MNIST) with just a single search space. This is not enough to support the claims from the abstract in this regard.
- Several important parts of the paper miss an in-depth explanation and instead refer the reader to the appendix, which hinders coherency.

**Potential Impact On The Field Of Automl:**

Reducing the runtime of NAS methods is one of the main goals of AutoML research and therefore of high importance to the community.

**Review Confidence:**

2

**Review Rating:**

6

**Review Summary:**

In summary, the method presented in this paper and most of the results are sound, but it is missing some clarity in its presentation.

**Technical Quality And Correctness:**

The Adaptive-DPT algorithm is derived by combining two methods: DARTS/DARTS-PT as a one-shot approach and GLISTER\GRAD-MATCH as the adaptive subset selection procedure. The theoretical derivations of both methods and the empirical evaluations of Adaptive-DPT that are presented in the main paper and the appendix are very detailed.

---

### Official Review · Reviewer_FWiR · 2024-03-28

**Potential Impact On The Field Of Automl Rating:** 4
**Technical Quality And Correctness:** The algorithms and experiments look c…
**Technical Quality And Correctness Rating:** 4
**Clarity Rating:** 3

**Summary Of Contributions:**

This paper combines subset selection of training datasets with both ero-shot NAS and black box algorithms for more efficient architecture search.

**Actions Required To Increase Overall Recommendation:**

-

**Clarity:**

The paper is written clearly and consistently, it's easy to follow and reasonably self-contained. Only the experiments section would benefit from a restructuring, ideally having the regularization paragraph with the methods and structuring the discussion a bit more (by e.g. separating the DEHB/BOHB results from the rest of the paragraph).

Minor details:
- brackets around citations are somewhat inconsistent with some citation blocks being placed without any brackets and some brackets not being opened or closed (as e.g. in line 21). In my opinion, it would be better to consistently either use "citep" to generate brackets automatically or not use any at all.
- similarly, some citations of specific methods are placed directly in the text without brackets. This is not great for readability and I'd recommend using "citep" as a standard command to avoid this.
- Tables 7 and 8 are terrible to read without better labels and descriptions. Yes, you refer to a different figure, but I still think they should be able to stand on their own.

**Overall Review:**

The combination of subset selection and NAS is very relevant in my opinion. The paper shows nicely how much faster NAS can be done with subset selection compared to using the full training set, whether using DARTS-PT or black box optimizers. Overall, I think it's a good paper with an important contribution even if the combination of two existing approaches is relatively "simple". I recommend accepting it.

Nevertheless, I have a few comments:
- I don't think it's correct to say that Adaptive-DPT has "the highest performance across all search spaces" (l. 231) - it's not true in Table 4 and generally there's often a tie. That is perfectly fine since performance improvements in terms of test accuracy are not the target of the paper anyway, so this sentence feels like an unnecessary overclaim.
- Do you have an intuition why on all search spaces but NAS-Bench 201 ImageNet, the random and entropy variants have DARTS-PT within their standard deviation? That seems incredibly good for this sort of sampling, does that mean we need to evaluate such methods on harder datasets?

**Potential Impact On The Field Of Automl:**

Being able to use only a fraction of training data would mean a significant performance upgrade across the field.

**Review Confidence:**

3

**Review Rating:**

8

**Review Summary:**

A good paper providing big compute time savings for NAS with surprisingly well-performing random baselines. Should be accepted in my opinion.

---

### Official Review · Reviewer_vm7G · 2024-03-28

**Potential Impact On The Field Of Automl Rating:** 3
**Technical Quality And Correctness Rating:** 1
**Clarity Rating:** 4

**Summary Of Contributions:**

This paper introduces an adaptive subset selection approach to neural architecture search (NAS), which significantly reduces the computational cost of leading NAS algorithms, including DARTS-PT, BOHB, and DEHB, while maintaining or even improving their performance.

**Actions Required To Increase Overall Recommendation:**

Please take into account my previous comments and consider implementing them to enhance the paper.

**Clarity:**

Overall, the paper is well-written and easy-to-follow. I have some minor comments that might improve the clarity of the paper:
- Related Work-> There exist related studies that have been missing from the literature review.
    - Prasad, V., White, C., Jain, P., Nayak, S., Iyer, R. K., & Ramakrishnan, G. (2022, May). Speeding up NAS with Adaptive Subset Selection. In First Conference on Automated Machine Learning (Late-Breaking Workshop).
    - Jain, Eeshaan, et al. "Efficient Data Subset Selection to Generalize Training Across Models: Transductive and Inductive Networks." Advances in Neural Information Processing Systems 36 (2024).
    - Yao, P., Liao, C., Jia, J., Tan, J., Chen, B., Song, C., & Zhang, D. (2023). ASP: Automatic Selection of Proxy dataset for efficient AutoML. arXiv preprint arXiv:2310.11478.

**Overall Review:**

# Strengths:
- Paper is well-written and addresses important research questions.
# Weaknesses:
- The novelty of the paper is under question.
    - Prasad, V., White, C., Jain, P., Nayak, S., Iyer, R. K., & Ramakrishnan, G. (2022, May). Speeding up NAS with Adaptive Subset Selection. In First Conference on Automated Machine Learning (Late-Breaking Workshop).
- Results are limited to CNN models. I would like to see if (and how) the proposed method could be extended to transformers.
- Compare the results of the proposed method with:
    - (Table 5) Jain, Eeshaan, et al. "Efficient Data Subset Selection to Generalize Training Across Models: Transductive and Inductive Networks." Advances in Neural Information Processing Systems 36 (2024).

**Potential Impact On The Field Of Automl:**

Green AutoML: this paper contributes to higher energy-efficiency of using NAS methods.

**Review Confidence:**

3

**Review Rating:**

4

**Review Summary:**

The paper would strongly benefit from extending results to
- transformers,
- evaluating memory and bandwidth usage of GPU, and
- comparison with SoTA methods published in 2023 and 2024.

**Technical Quality And Correctness:**

Overall, the approach and reported results are technically correct and sound. HOWEVER, a substantially similar paper with the same title and results has been published in AutoML 2022. This raises questions about whether this work is an extension of the previous research. Additionally, the absence of citation for the earlier paper is notable. Although the method names have been changed in this submission ("Adaptive-NAS" to "Adaptive-Dpt"), the exact results are reported in tables, suggesting that this may not be an extension of the previous paper. I eagerly await further communication from the authors to address these concerns and potentially update my evaluation accordingly.

- Prasad, V., White, C., Jain, P., Nayak, S., Iyer, R. K., & Ramakrishnan, G. (2022, May). Speeding up NAS with Adaptive Subset Selection. In First Conference on Automated Machine Learning (Late-Breaking Workshop).

---

### Official Review · Reviewer_esJ4 · 2024-04-02

**Potential Impact On The Field Of Automl Rating:** 3
**Technical Quality And Correctness Rating:** 4
**Clarity:** The paper is written clearly.
**Clarity Rating:** 3

**Summary Of Contributions:**

This paper introduces a novel approach to Neural Architecture Search (NAS) by integrating adaptive subset selection, significantly reducing the computational cost of finding high-performing architectures without sacrificing accuracy. The key contributions include the development of Adaptive-Dpt, the first NAS algorithm employing adaptive subset selection, and the extension of this concept to enhance hyperparameter optimization algorithms, resulting in Adaptive-BOHB and Adaptive-DEHB. These methods demonstrate substantial runtime reductions across multiple datasets and search spaces while maintaining or improving accuracy. Furthermore, the paper promotes research reproducibility by releasing all experimental codes and introduces a comprehensive codebase that integrates various subset selection algorithms into one-shot NAS. The work opens up new directions for future research in NAS and presents broader implications for the development of efficient architecture search methods.

**Actions Required To Increase Overall Recommendation:**

To enhance the paper and potentially increase its recommendation, the authors could undertake several actions. Expanding the comparative analysis to include a broader spectrum of state-of-the-art NAS algorithms would provide a clearer benchmarking of the proposed method's performance and efficiency.

**Overall Review:**

Strength:
* The paper's core strength lies in its novel integration of adaptive subset selection with Neural Architecture Search (NAS), tackling the critical issue of computational efficiency.
* The paper did extensive experiments across multiple datasets, providing a robust evaluation of the effectiveness.

Weakness:
* The paper's main baseline is DARTS. It's better to include other NAS algorithms to make the baseline more comprehensive.

**Potential Impact On The Field Of Automl:**

This paper's contributions to integrating adaptive subset selection with Neural Architecture Search (NAS) could significantly impact the AutoML field, addressing the computational efficiency challenge without compromising model performance. The introduction of Adaptive-Dpt, along with enhancements to hyperparameter optimization through Adaptive-BOHB and Adaptive-DEHB, showcases notable advancements in reducing the runtime and computational resources required for effective model search.

**Review Confidence:**

2

**Review Rating:**

6

**Review Summary:**

This paper presents an innovative approach by integrating adaptive subset selection into Neural Architecture Search (NAS), addressing the crucial issue of computational efficiency in AutoML. Its strength lies in the comprehensive experimental validation across multiple datasets, showcasing the method's effectiveness. However, the study's reliance on DARTS as the primary baseline limits the breadth of its comparative analysis. Including a wider array of NAS algorithms could enhance the evaluation, offering a clearer perspective on the proposed method's relative performance. Expanding the comparison would underscore the innovation and efficiency of the approach, solidifying its contributions to the NAS and AutoML fields.

**Technical Quality And Correctness:**

The paper combines adaptive subset selection with Neural Architecture Search (NAS) to enhance efficiency without sacrificing model performance, showing high technical quality. Its innovative solution addresses AutoML's computational challenge with thorough experiments across diverse datasets, demonstrating its effectiveness.

---

### Meta-Review · Area_Chair_kcRZ · 2024-04-22

**Paper Recommendation:** Accept
**Confidence:** 5

**Metareview:**

This paper is a follow-up of earlier work presented at the AutoML workshop 2022. The extension is quite considerable and well executed. The reviewers acknowledge the good fit to the conference. It's a clear case in my opinion, the paper should be accepted to the main track. I thank reviewers and authors for their constructive discussions.

---

### Decision · Program_Chairs · 2024-04-29

**Decision:**

Accept

**Comment:**

Thank you for submitting your paper. We are happy to tell you that we accept your paper to the main track. See you in Paris.